# Quality, productivity, and economic implications of exoskeletons for occupational use: A systematic review

**Daniel E. Fournier**[1], **Marcus Yung**[1], **Kumara G. Somasundram**[1], **Bronson B. Du**[1], **Sara Rezvani**[1], **Amin Yazdani**[1,2]*

1 Canadian Institute for Safety, Wellness & Performance, School of Business, Conestoga College Institute of Technology and Advanced Learning, Ontario, Canada, 2 School of Public Health and Health Systems, University of Waterloo, Ontario, Canada

☯ These authors contributed equally to this work.
* ayazdani@conestogac.on.ca

**Data Availability Statement:** All relevant data are within the paper and its Supporting Information files.

**Funding:** The research was co-funded by the Social Sciences and Humanities Research Council

## Abstract

The objective of this systematic review was to synthesize the current state of knowledge on the quality and productivity of workers and their work while wearing exoskeletons, as well as the economic implications of exoskeletons for occupational use. Following the PRISMA guidelines, six databases were systematically searched for relevant journal articles, written in English, and published since January 2000. Articles meeting the inclusion criteria had their quality assessed using JBI's Checklist for Quasi-Experimental Studies (Non-Randomized Experimental Studies). A total of 6,722 articles were identified and 15 articles focusing on the impact of exoskeletons on quality and productivity of exoskeleton users while performing occupational tasks were included in this study. None of the included articles evaluated the economic implications of exoskeletons for occupational use. This study revealed several quality and productivity measures (e.g., endurance time, task completion time, number of errors, number of task cycles completed) used to evaluate the impact of exoskeletons. The current state of the literature suggests that quality and productivity impacts of exoskeleton use are dependent on task characteristics that should be considered when adopting exoskeletons. Future studies should evaluate the impact of exoskeleton use in the field and on a diverse pool of workers, as well as its economic implications to better support decision-making in the adoption of exoskeletons within organizations.

## 1. Introduction

An exoskeleton is defined as "a wearable device that augments, enables, assists, or enhances motion, posture, or physical activity" [1]. Exoskeletons have been used for occupational [2] and rehabilitation [3] purposes. Exoskeletons can be classified by type (i.e., active or passive) and by which body part is being supported (i.e., lower body, upper body, full-body) [4]. Active exoskeletons consist of external actuators such as electric motors, hydraulic actuators, and pneumatic muscles to augment the user's power and provide extra energy [4, 5]. In contrast,

(SSHRC) and through Natural Sciences and Engineering Research Council under College and Community Social Innovation Fund. The funders had no role in study design, data collection and analysis, decision to publish, or preparation of the manuscript. Initials of author who received the award: AY, MY SSHRC Ref.: 970-2021-1007 Funding Agency Website: https://www.sshrc-crsh.gc.ca/home-accueil-eng.aspx.

**Competing interests:** The authors have declared that no competing interests exist.

passive exoskeletons do not use any actuators. Instead, they rely on elements such as springs and dampers to store and release energy generated by the user's movements to support a posture or motion at a specific joint [4, 5]. Several previous systematic reviews have provided a comprehensive review on the efficacy of different types of exoskeletons in both industrial applications (e.g., prevention of workplace injuries) and rehabilitative purposes (e.g., functional mobility) [2, 3, 6–8].

In recent years, there has been a large focus on the development of exoskeletons as a solution to decrease the risk of musculoskeletal disorders (MSD) in the workplace [9, 10]. Work-related MSD, including sprains and strains, are among the most common injuries and represent more than 30% of total claims in Ontario, Canada [11]. Previous research suggests that the use of exoskeletons may mitigate MSD risk by reducing muscle loading and physical stress and strain in work-related tasks [4, 5]. For example, wearing an exoskeleton during a lifting task has been shown to increase metabolic efficiency as well as decrease back muscle activation and low back loading [12].

Despite evidence that generally support the use of exoskeleton for reducing physical exposure and mitigating MSD risk, the health benefits alone may not necessarily facilitate its adoption in organizations. MSD prevention is often treated as an organizational occupational health and safety "side-car", with inadequate resources that restrict its application [13–16]. Hence, ergonomists and/or occupational health and safety professionals may need to spend considerable time to gain credibility and obtain support to implement change [14, 17]. On the other hand, concepts such as quality, productivity, and cost are powerful business agendas [18, 19] that would likely receive more resources and attention. Research has shown that the adoption of exoskeletons is primarily influenced by their impacts on quality and productivity, as well as their economic implications, rather than their potential to reduce MSD risks alone [20, 21]. Exoskeletons are more likely to be adopted if they demonstrate higher impact on quality and productivity [20]; however, they are less likely to be adopted if they are too expensive and have low return on investment [21]. As a result, understanding the impacts of exoskeleton on quality and productivity, as well as its economic implications, can support decision-making in the adoption of exoskeletons in organizations.

To date, no systematic reviews have examined the quality, productivity, and economic impacts of exoskeleton use in occupational tasks. McFarland & Fischer [22] conducted a systematic review of the effects of upper limb exoskeletons on physical exposures. The authors reported on quality and productivity impacts of exoskeleton use; however, this topic was not the primary focus of their review, and did not explicitly include search terms related to quality and productivity. Given the current gaps in the literature, there is a need for more research that directly sheds light on this important topic. Therefore, the objective of our systematic review was to synthesize the current state of knowledge on the quality and productivity of workers and their work while wearing exoskeletons, as well as the economic implications of exoskeletons for occupational use.

## 2. Methods

### 2.1. Search strategy

A systematic review was conducted following the Preferred Reporting Items for Systematic Reviews and Meta-Analyses (PRISMA) guidelines to identify, review and extract data from journal articles [23]. In consultation with research librarians, a list of key words was developed for each of the three concepts: 1) exoskeletons; 2) work and occupation; and 3) quality, productivity, and economics (Table 1). Quality, productivity, and economic terms were adapted from an initial key word list from Hackney et al. [24], which included terms that describe

**Table 1. Search terms used for electronic literature search.**

| Concept | Search Terms |
|---|---|
| Exoskeleton | exoskeleton, exoskeleton device, assistive technology, weight bearing, wearable, device, arm, shoulder, lower body, back, whole body, trunk, support, robotic* |
| Work & Occupation | work*, occupation*, employee*, labour* |
| Quality, Productivity, and Economics | work performance, value, benefit*, cost*, effectiveness, claim*, economic evaluation, operating, job loss, productivity, efficiency, investment, performance, monetary, non-monetary, profit*, return on investment, net present value, job satisfaction, presenteeism, absenteeism, quality, task completion, turnover, impact |

*Denotes terms that were searched using all possible suffixes.

measurable outcomes. The Boolean operators "OR" and "AND" were used between search terms within a concept and across concepts respectively. When available, subject headings for their respective databases were also used in the literature search. An electronic literature search was performed using six databases: PubMed, Medline, Embase, Business Source Complete, IEEE Xplore, and Scopus. The combination of the selected databases helped ensure articles published in scientific journals pertaining to the application of exoskeletons in the medical-, occupational-, engineering-, and business-related fields were captured. Search results were filtered for peer-reviewed journal articles, written in the English language, and published since January 2000. A total of 6,052 articles were identified for screening after 670 duplicates were removed (Fig 1).

## 2.2. Selection criteria

We included original research articles that used exoskeleton in occupational tasks and evaluated its effects on quality or productivity of workers and their work, or its economic impacts. For example, we included articles that studied how exoskeletons affected occupational task completion time. Articles were excluded for the following reasons: (a) article was a review paper, (b) article was written in language other than English, (c) exoskeleton was not used in the study, (d) exoskeleton was used for rehabilitation, (e) article measured quality metrics (e.g., electromyography, range of motion, etc.) without relating them to productivity and/or quality, (f) exoskeleton use was not related to a work setting or occupational tasks, and (g) exoskeleton use was not related to quality, productivity, or economic impacts.

## 2.3. Screening tool

We designed and implemented a screening tool based on the inclusion and exclusion criteria. Titles and abstracts of 300 randomly selected articles were independently screened by five reviewers (DF, BD, KS, MY, AY) to ensure inter-rater reliability with the use of the screening tool. Any discrepancies in the use of the screening tool for inclusion and exclusion between reviewers were discussed until consensus was achieved. Following these discussions, the screening tool was refined to ensure consistent application. Title and abstracts of an additional 30 randomly selected articles were screened by all five reviewers to re-assess the inter-rater reliability of the revised screening tool [25]. After this round of screening, substantial agreement between the five raters was achieved (Fleiss' kappa value of K = 0.69 [25]). The screening tool is presented in S1 Appendix.

## 2.4. Title and abstract screening & full-text review

Given substantial reliability with the screening tool for the title and abstract screening (330 articles), the remaining 5722 articles were divided and independently screened by five

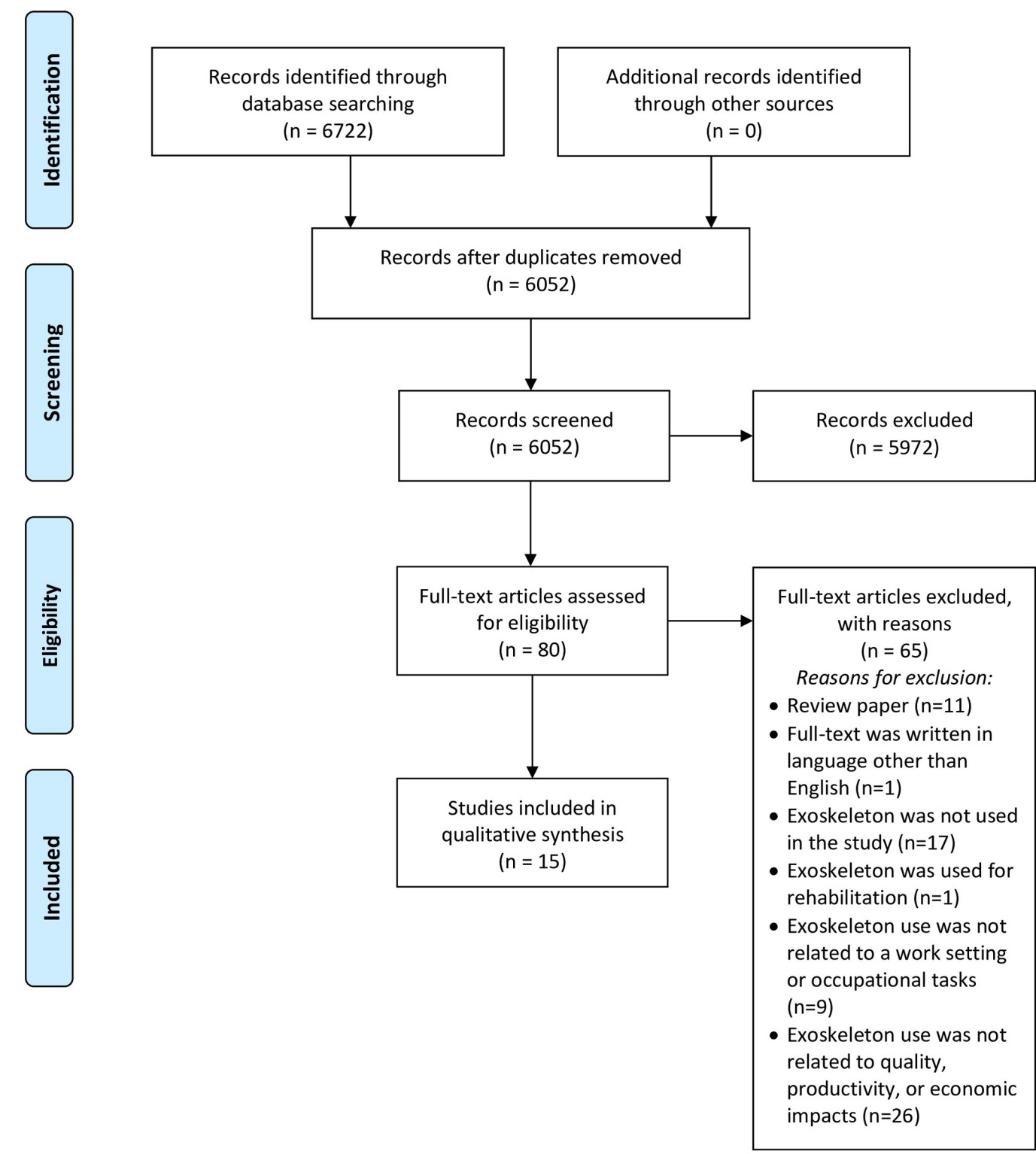

From: Moher D, Liberati A, Tetzlaff J, Altman DG, The PRISMA Group (2009). *P*referred *R*eporting *I*tems for *S*ystematic Reviews and *M*eta-*A*nalyses: The PRISMA Statement. PLoS Med 6(7): e1000097. doi:10.1371/journal.pmed1000097

**For more information, visit www.prisma-statement.org.**

**Fig 1. PRISMA flow diagram of literature search and screening process for selected articles.**

reviewers based on their titles and abstracts using Covidence software [26]. When a reviewer was unsure of the relevancy of the article, the article was retained for full-text review. Eighty articles were retained after the title and abstract screening and were reviewed in full independently by two reviewers (DF, SR). Any discrepancies between reviewers for inclusion of an article were discussed by both reviewers until consensus was reached. A total of 15 articles were included in the final review and 65 studies were excluded due to: being a review paper (n = 11), full-text written in language other than English (n = 1), not using an exoskeleton (n = 17), using an exoskeleton for rehabilitation (n = 1), not linking exoskeleton use to a work setting or occupational tasks (n = 9), and not linking exoskeleton use to quality, productivity, or economic impacts (n = 26).

### 2.5. Quality appraisal & data extraction

Two reviewers (DF, SR), who completed the full-text review, independently assessed the quality of the 15 articles included in the final review using the JBI Critical Appraisal Tools [27]. JBI Checklist for Quasi-Experimental Studies (Non-Randomized Experimental Studies) included nine questions and was used to appraise quasi-experimental studies [27]. The research team determined a cut off value of 6, a priori, as an acceptable score to assess the articles as good quality [28, 29]. The score of each article was calculated based on the number of components included from the checklist. A consensus-based decision was reached by the two reviewers for each article and any disagreement was resolved through discussion. Following the quality appraisal, data was extracted from each article using a custom data extraction tool in Covidence.

## 3. Results

We identified 15 articles pertaining to the impact of exoskeletons on quality and productivity of workers and their work. Quality and productivity were evaluated in each study using objective measures. No articles assessed the economic impact (e.g., cost, return on investment) of exoskeletons.

Extracted data included: characteristics of each article (i.e., author, year, country, study design, participant description, and quality appraisal score) (Table 2); exoskeleton characteristics, occupational tasks performed, as well as quality and productivity measures and results (Table 3). Fifteen unique brands and types of exoskeletons were described in the included articles, consisting of 1 active exoskeleton, 12 passive exoskeletons, 1 wearable robot suit, and 1 exoskeleton where the type was not reported. Thirteen studies evaluated exoskeletons in a laboratory setting and simulated tasks such as drilling, manual material handling, manual assembly, and military operations. The remaining two studies evaluated exoskeleton use in a realistic setting such as snow shoveling, welding, and electrostatic painting. All articles received a score of 6 or more on the Checklist for Quasi-Experimental Studies (Non-Randomized Experimental Studies), indicating good quality.

### 3.1. Quality

Five studies assessed the impacts of exoskeleton on quality measures in occupational tasks such as simulated drilling (n = 3), simulated military operations (n = 1), and welding and electrostatic painting (n = 1) (Table 3). Metrics of quality included number of errors (n = 2), precision (n = 1), reaction time (n = 1), number of misses (n = 1), as well as other metrics specific to welding and electrostatic painting (n = 1) (Table 3).

There were mixed findings surrounding the effects of exoskeleton on drilling quality. Impact of exoskeleton on number of errors during repetitive drilling tasks has been shown to

**Table 2. Article characteristics of included research papers (sorted by year).**

| Author, Year | Country | Study Design | Participant Sample Size and Description | Quality Appraisal Score |
|---|---|---|---|---|
| Bosch et al., 2016 [30] | Netherlands | Repeated measures laboratory study | 18 (9M, 9F) Healthy Adults | 8 |
| Butler, 2016 [31] | USA | Repeated measures field study | 4[a] (2 Welders, 2 Painters) | 6 |
| Kim et al., 2018 [9] | USA | Repeated measures laboratory study | 12 (6M, 6F) University Students & Community Members | 7 |
| Miura et al., 2018 [32] | Japan | Repeated measures field study | 9 All Healthy Males | 7 |
| Miura et al., 2018 [33] | Japan | Repeated measures laboratory study | 18 (11M, 7F) Healthy Adults | 8 |
| Alabdulkarim et al., 2019 [34] | USA | Repeated measures laboratory study | 12 (7M, 5F) University Students & Community Members | 8 |
| Bequette et al., 2020 [35] | USA | Repeated measures laboratory study | 12 Military, All Healthy Males | 8 |
| Gruevski et al., 2020 [36] | Canada | Repeated measures laboratory study | 2 Canadian Infantry Regular Force Unit, All Healthy Males | 8 |
| Madinei et al., 2020 [37] | USA | Repeated measures laboratory study | 18 (9M, 9F) University Students & Community Members | 8 |
| Maurice et al., 2020 [38] | Slovenia | Repeated measures laboratory study | 12 College Students, All Healthy Males | 8 |
| Luger et al., 2021 [39] | Germany | Repeated measures laboratory study | 36 (2 excluded) All Healthy Males | 8 |
| Ogunseiju et al., 2022 [40] | USA | Repeated measures laboratory study | 10[a] Students | 8 |
| De Bock et al., 2022 [10] | Belgium | Repeated measures laboratory study | 22 All Healthy Males | 8 |
| Garosi et al., 2022 [41] | Iran | Repeated measures laboratory study | 14 University Students, All Healthy Males | 8 |
| Pinho & Forner-Cordero, 2022 [42] | Brazil | Repeated measures laboratory study | 14 (12M, 2F) Automotive Industry Workers | 8 |

Note: M = males; F = females

[a] Distribution of sex was not specified.

depend on work height [9] and design of exoskeleton [34]. When using the exoskeleton in a drilling task, the number of errors increased at overhead height but did not significantly change at shoulder height [9]. Number of errors in overhead drilling increased with the use of a passive upper extremity exoskeleton or a passive full-body exoskeleton; however, there was no significant change in errors when using a different passive upper extremity exoskeleton [34]. Exoskeleton use during overhead drilling did not significantly affect drilling force precision [10].

Positive effects were reported for quality in welding and electrostatic painting when using an exoskeleton [31]. Welders who used an exoskeleton experienced quality improvements in their task when considering several metrics including position, work angle, and travel angle. Exoskeleton use also resulted in greater quality in electrostatic painting based on measures such as visual defects (light pain and runs), film thickness, and dry thickness.

Exoskeleton use demonstrated mixed impacts on quality in a military obstacle course simulation that included visual and auditory tasks [35]. Reaction time in the visual task (i.e., respond to light targets by pressing a button on a simulated rifle) increased in some participants when using a powered exoskeleton. Similarly, reaction time in the auditory task (i.e., answering simulated radio calls) increased in all participants with the powered exoskeleton. Exoskeleton use, however, did not significantly affect the number of misses in both tasks.

**Table 3. Exoskeleton brand and type, occupational tasks performed, as well as quality and performance measures and results (sorted by year).**

| Author, Year | Exoskeleton Brand | Exoskeleton Type | Occupational Tasks Performed | Quality & Productivity Measures | Quality & Productivity Results |
|---|---|---|---|---|---|
| Bosch et al., 2016 [30] | Laevo | Passive exoskeleton | Static trunk forward flexion with both arms hanging down vertically | Endurance time (min) | ↑ endurance time |
| Butler, 2016 [31] | "Personal Ergonomic Device" (PED) | Not reported | • Electrostatic Painting<br>• Welding | • Electrostatic Painting: light pain and runs, film thickness, transfer efficiency, dry thickness, and time to paint<br>• Welding: position, contact tip work distance, work angle, travel angle and travel speed, and total weld time | ↑ quality and productivity<br>Note: no statistical tests were conducted to detect significant differences |
| Kim et al., 2018 [9] | EksoVest (prototype) | Passive upper extremity exoskeletal vest | • Repetitive drilling task at different work heights (overhead and shoulder heights)<br>• Light assembly task (wiring task) at different work heights | • Number of errors<br>• Task completion time (s) | • ↑ number of errors in overhead drilling<br>• ↓ drilling task completion time, regardless of work height<br>• Ø wiring task completion time |
| Miura et al., 2018 [32] | HAL–hybrid assisted limb for Care Support | Wearable robot suit for lumbar support | Snow-shoveling | • Number of scoops<br>• Shoveling endurance time (s)<br>• Shoveling distance (m) | ↑ number of scoops, shoveling endurance time, and shoveling distance |
| Miura et al., 2018 [33] | HAL–hybrid assisted limb for Care Support | Wearable robot suit for lumbar support | Repetitive lifting task | • Number of lifts<br>• Lifting endurance time (s) | ↑ number of lifts and lifting endurance time |
| Alabdulkarim et al., 2019 [34] | 1. Exovest<br>2. EksoWorks 3. FORTIS | 1. Passive upper extremity exoskeleton vest<br>2. Passive upper extremity exoskeleton vest<br>3. Passive full-body exoskeleton | Overhead repetitive drilling task | Number of errors | ↑ number of errors when using the Exovest and FORTIS exoskeletons |
| Bequette et al., 2020 [35] | "Lower-extremity exoskeleton" (10kg) | Active lower-extremity exoskeleton | • Visual task: respond to light targets by pressing a button on a simulated rifle<br>• Auditory task: answering simulated radio calls<br>• Follow task: follow researcher at a specified distance | • Visual task: number of misses and visual reaction time (s)<br>• Auditory task: number of misses and auditory reaction time (s)<br>• Follow task: incremental lag time (s) | • ↑ reaction time in visual task with powered exoskeleton than without exoskeleton for 5 of 12 participants<br>• ↑ reaction time in auditory task with powered exoskeleton than unpowered exoskeleton for all participants<br>• Ø number of misses in visual and auditory tasks<br>• ↑ incremental lag time in follow task with unpowered exoskeleton than powered/without exoskeleton for 1 participant |
| Gruevski et al., 2020 [36] | UPRISE Gen 3.0 (customized prototype) | Passive full-body exoskeleton | Can-LEAP obstacle course, that consists of 10 obstacles such as sprint, agility run, crawls, casualty drag, etc | Task completion time (s) | • ↑ total obstacle course completion time<br>• Ø completion time for running or carrying tasks<br>• ↑ completion time for tasks with confined spaces<br>Note: no statistical tests were conducted to detect significant differences |
| Madinei et al., 2020 [37] | 1. BackX (model AC)<br>2. Laevo V2.5 | 1. Passive back-support exoskeletons<br>2. Passive back-support exoskeleton | Precision manual assembly task (insert pegs in a pegboard) in seated and unseated positions | Completion time (s) | • ↑ task completion time with the use of BackX and Laevo for females during seated and unseated positions<br>• ↑ task completion time with the use of BackX for males during unseated position |

*(Continued)*

**Table 3.** (Continued)

| Author, Year | Exoskeleton Brand | Exoskeleton Type | Occupational Tasks Performed | Quality & Productivity Measures | Quality & Productivity Results |
|---|---|---|---|---|---|
| Maurice et al., 2020 [38] | PAEXO | Passive upper-limb exoskeleton | Overhead pointing task with a power drill | Duration of movement (s) | Ø movement duration |
| Luger et al., 2021 [39] | Laevo V2.56 (2.8kg) | Passive back-support exoskeleton | • Pallet box lifting<br>• Fastening screws<br>• Lattice box lifting | Time-to-task-accomplishment (s) | ↑ time-to-task-accomplishment for pallet box lifting and lattice box lifting<br>Ø time-to-task-accomplishment for fastening screws |
| Ogunseiju et al., 2022 [40] | FLx ErgoSkeleton | Passive postural-assist exoskeleton | Manual material handling tasks: lifting, moving, and placing wooden planks | Completion time (s) | Ø task completion time |
| De Bock et al., 2022 [10] | Exo4Work | Passive cable-driven shoulder exoskeleton | • Wiring<br>• Drilling<br>• Lifting | • Number of wires connected<br>• Drilling force precision (N)<br>• Lifting task completion time (s) | • Ø number of wires connected and drilling force precision<br>• ↑ completion time of one lifting cycle |
| Garosi et al., 2022 [41] | "Head/neck supporting exoskeleton" (HNSE) | Passive head/neck supporting exoskeleton | Repetitive fastening/unfastening nut task at overhead work height | Number of fastened/unfastened nuts | Ø number of nuts fastened/unfastened |
| Pinho & Forner-Cordero, 2022 [42] | ShoulderX (V1) | Passive upper-limb exoskeleton | Manual screwing/unscrewing task at different work heights | Task completion time (s) | ↑ task completion time at shoulder height |

Note: ↑ or ↓ = increase or decrease with exoskeleton use compared to without exoskeleton; Ø no difference between with and without exoskeleton. Results are statistically significant unless otherwise specified.

## 3.2. Productivity

Thirteen studies examined the effects of exoskeleton on productivity in occupational tasks such as welding and electrostatic painting (n = 1); simulated military operations (n = 2); shoveling (n = 1); as well as simulated manual material handling, drilling, and/or manual assembly tasks (n = 9) (Table 3). Productivity metrics included task completion time (n = 8); endurance time (n = 3); movement duration (n = 1); number of task cycles completed (n = 4); and other metrics specific to shoveling (n = 1), military operations (n = 1), as well as welding and electrostatic painting (n = 1) (Table 3).

Exoskeleton use positively affected productivity in shoveling as well as welding and electrostatic painting. Number of scoops, shoveling endurance time, and shoveling distance increased with exoskeleton use [32]. When using an exoskeleton, welders demonstrated higher productivity according to metrics such as contact tip work distance, travel speed, and total weld time [31]. Similarly, exoskeleton use improved productivity in electrostatic painting based on measures such as transfer efficiency and time to paint.

There were mixed findings regarding the effects of exoskeleton on productivity in military obstacle course simulations. When using a passive full-body exoskeleton, completion time increased in tasks with confined spaces but did not significantly change in running or carrying tasks during the simulation [36]. Additionally, when military personnel were instructed to follow a researcher at a specified distance, only one of 12 participants demonstrated longer incremental lag time with the use of an unpowered lower-extremity exoskeleton [35].

The literature revealed mixed evidence on the productivity impacts of exoskeleton in simulated manual material handling, drilling, and manual assembly tasks. Lifting endurance time

and number of lifts in a repetitive lifting task increased with exoskeleton use [33]; however, task completion time was either longer [10, 39] or did not significantly change [40] when using exoskeletons. Exoskeleton use has been shown to reduce completion time in a drilling task, regardless of work height [9]; however, it did not significantly affect movement duration during an overhead pointing task with a power drill [38]. Using an exoskeleton in manual assembly tasks, such as connecting wires and fastening nuts or screws, did not significantly alter task completion time [9, 39] and number of task cycles completed [10, 41]; however, it negatively affected completion time in a manual screwing task at shoulder height [42]. Interestingly, one study demonstrated that the impact of two passive back-support exoskeletons on completion time in a simulated manual assembly task (i.e., inserting pegs in a pegboard) was dependent on the sex of the participant during different seating conditions [37]. Madinei et al. (2020) [37] found that task completion time increased for females when using either of the two passive back-support exoskeletons in seated and unseated positions, whereas for males, only one of the exoskeletons resulted in a longer completion time in unseated positions. A passive back-support exoskeleton has also been shown to improve endurance time when participants were instructed to maintain a forward trunk flexion in a simulated manual assembly setup [30].

## 4. Discussion

In this systematic review, we synthesized the current state of knowledge on the quality and productivity of workers and their work while wearing exoskeletons, as well as the economic implications of exoskeletons for occupational use. Based on 15 articles, we observed mixed evidence in the current literature regarding the quality and productivity impacts of exoskeleton use. None of the included articles assessed the economic impacts of exoskeletons.

There was mixed evidence regarding the quality and productivity impacts of exoskeleton use reported in the analyzed literature; this finding was irrespective of the type of exoskeleton (passive vs. active), the brand of exoskeleton, and the supported body part(s). Although the literature reported mixed results across all types and brands of exoskeletons, their effectiveness was dependent on task characteristics (e.g., static vs. dynamic movements, workspace, etc.). For example, using a passive exoskeleton increased endurance time for tasks requiring static postures like forward trunk flexion to be held [30]. However, the same exoskeleton would potentially not be as effective for dynamic tasks such as manual material handling due to its negative impact on task completion time [39].

None of the articles evaluated the economic implications of exoskeletons for occupational use. Studies analyzing economic impacts of occupational health and safety interventions in the workplace have been generally rare [43, 44]. The lack of studies may be due to workplace challenges (e.g., insufficient financial data available from organizations, conflicting priorities among stakeholders, etc.) and limited expertise in economic analysis among health and safety researchers [43]. Economic impacts of exoskeletons can have a strong influence on their implementation in organizations [21]. Developing resources based on research evidence in peer reviewed publications on the financial implications of exoskeletons will provide crucial evidence and data for the adoption and uptake of these technologies by organizations.

The current state of the literature also places an emphasis on lab-based studies in comparison to field studies. Out of the 15 articles assessed, only two [31, 32] evaluated the effects of exoskeleton on quality and productivity in the field. Job dynamics in actual work environments are more complex than lab settings [45]. Factors such as safety and working conditions (e.g., personal protective equipment, weather condition, confined spaces) may also influence the effectiveness of exoskeletons in the field and cannot be fully replicated in a lab environment [37].

Out of the 15 articles evaluated in this systematic review, only five studies [9, 30, 33, 34, 37] included women as part of the study population, and out of the five, only two articles [34, 37] analyzed gender differences. Lack of a diverse study population in most of the included articles may limit the generalizability of their results. Sex and gender differences were not often considered likely because the majority of occupations that may potentially benefit from exoskeleton implementation were male-dominated jobs [46]. However, in many sectors including the skilled trades, there is a significant focus on diversifying the workforce to counter the current nationwide shortage of skilled labour [47].

## 4.1. Recommendations for future studies

Based on the findings of our systematic review, we propose several recommendations for future studies. Future research on exoskeletons should focus on assessing quality and productivity of workers as we only found 15 articles; most importantly, as our research did not yield articles evaluating the economic implication of exoskeletons, future research should focus on cost-benefit analysis and return on investment to justify the benefits of exoskeleton adoption for the organization and its workers. Evaluation studies have been primarily lab-based, limiting their ecological validity; we encourage researchers to evaluate exoskeletons with actual workers while completing their job tasks on worksites. Furthermore, the participants in future research studies should reflect the emerging diverse workforce including women and people from underrepresented groups. Without more information on the impacts of exoskeleton use in real work environments and on a diverse work population as well as its economic implications, these gaps in the literature will continue to hinder organizations from adopting exoskeletons and may limit the full potential and application of these technologies.

## 4.2. Limitations

There are several limitations to consider for this review. First, literature that may have included some aspects of quality and productivity described above may have been excluded during the screening process, due to poor descriptions in the titles and abstracts. In order to capture all potential articles, the reviewers included articles they were unsure of for full-text review if the title and abstract did not lead to a conclusive decision. Second, literature prior to January 2000 and conference papers were excluded since the inclusion criteria required original peer-reviewed journal articles. However, this is a common limitation as conference abstracts often contain information that may be inadequate and not dependable for inclusion [48]. Third, our literature search was based on an a priori list of selected key words and subject heading terms compiled after consulting with research librarians; however, we may have missed certain search key terms. Different groups assessing the same research question may have arrived at a different list of key words and subject heading terms that may have led to different search results from each database. However, the generated key word list for this systematic review was compiled after a consensus was reached after numerous discussions between the team of authors and librarians. Given the considerable body of evidence from the search, this common limitation would not have affected the results.

## 4.3. Conclusion

Our systematic review provided mixed evidence regarding the effectiveness of exoskeleton based on quality and productivity measures. We also observed an absence of evidence on the economic impacts (e.g., cost, return on investment) of exoskeletons for occupational use. When selecting and adopting exoskeletons, task characteristics (e.g., required movements, workspace) may need to be considered to achieve the most effective outcomes for quality and

productivity of workers and their work. More empirical studies are needed to improve our understanding on quality, productivity, and economic impacts of exoskeletons, including studies that consider sex and gender as well as studies that take place in actual work environments.

## Supporting information

**S1 Checklist. PRISMA checklist.**
(DOC)

**S1 Appendix. Screening tool.**
(DOCX)

**S1 Dataset. Minimal dataset.**
(CSV)

## Acknowledgments

We would like to acknowledge Danya Goldsmith Milne and Juliet Conlon for their assistance and expertise in compiling the list of key search terms used for this systematic review.

## Author Contributions

**Conceptualization:** Marcus Yung, Bronson B. Du, Amin Yazdani.

**Data curation:** Daniel E. Fournier, Marcus Yung, Kumara G. Somasundram, Bronson B. Du, Sara Rezvani, Amin Yazdani.

**Formal analysis:** Daniel E. Fournier, Marcus Yung, Kumara G. Somasundram, Bronson B. Du, Sara Rezvani, Amin Yazdani.

**Funding acquisition:** Marcus Yung, Amin Yazdani.

**Investigation:** Daniel E. Fournier, Marcus Yung, Kumara G. Somasundram, Bronson B. Du, Sara Rezvani, Amin Yazdani.

**Methodology:** Daniel E. Fournier, Marcus Yung, Kumara G. Somasundram, Bronson B. Du, Amin Yazdani.

**Project administration:** Marcus Yung, Amin Yazdani.

**Resources:** Marcus Yung, Amin Yazdani.

**Software:** Daniel E. Fournier, Marcus Yung, Kumara G. Somasundram, Bronson B. Du, Sara Rezvani, Amin Yazdani.

**Supervision:** Marcus Yung, Amin Yazdani.

**Validation:** Daniel E. Fournier, Marcus Yung, Kumara G. Somasundram, Bronson B. Du, Sara Rezvani, Amin Yazdani.

**Visualization:** Daniel E. Fournier, Marcus Yung, Kumara G. Somasundram, Bronson B. Du, Sara Rezvani, Amin Yazdani.

**Writing – original draft:** Daniel E. Fournier.

**Writing – review & editing:** Daniel E. Fournier, Marcus Yung, Kumara G. Somasundram, Bronson B. Du, Sara Rezvani, Amin Yazdani.

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
