## [Decision Letter · Decision Letter 0]

9 May 2023

PONE-D-23-09205Effects of exoskeleton use on quality and productivity: A systematic reviewPLOS ONE

Dear Dr. Yazdani,

Thank you for submitting your manuscript to PLOS ONE. After careful consideration, we feel that it has merit but does not fully meet PLOS ONE’s publication criteria as it currently stands. Therefore, we invite you to submit a revised version of the manuscript that addresses the points raised during the review process.

 Associate Editor: Reviewers have raised some concerns that should be addressed. 

We look forward to receiving your revised manuscript.

Kind regards,

Noman Naseer, PhD

Academic Editor

PLOS ONE

Journal Requirements:

2. Please note that in order to use the direct billing option the corresponding author must be affiliated with the chosen institute. Please either amend your manuscript to change the affiliation or corresponding author, or email us at plosone@plos.org with a request to remove this option.

Additional Editor Comments:

Associate Editor: Reviewers have raised some concerns that should be addressed.

Reviewers' comments:

Reviewer's Responses to Questions

**Comments to the Author**

1. Is the manuscript technically sound, and do the data support the conclusions?

Reviewer #1: No

Reviewer #2: Yes

2. Has the statistical analysis been performed appropriately and rigorously? 

Reviewer #1: No

Reviewer #2: N/A

3. Have the authors made all data underlying the findings in their manuscript fully available?

Reviewer #1: Yes

Reviewer #2: Yes

4. Is the manuscript presented in an intelligible fashion and written in standard English?

Reviewer #1: Yes

Reviewer #2: Yes

5. Review Comments to the Author

Reviewer #1: In this manuscript authors analyzed the effects of exoskeleton use on Quality and Productivity. For this purpose 15 critical studies were selected and reviewed for the factors/measures of endurance time, task completion time, number of errors, number of task cycles completed etc. Following are the concerns and suggestion to improve the manuscript

1. As per my understanding, the author's intension was to measure the quality and productivity of produced, developed and available exoskeleton in and out of lab applications. For this purpose keywords selected were not focused and had deficiencies, like human power augmentation, HCI based exoskeleton, BCI based exoskeletons, neurorobotics etc. These keyword may provide other critical studies for the mentioned topic.

2. The Introduction section is very brief and lacks a comprehensive review including introduction of exoskeleton, their types and applications. This will help to better contextualize your study and highlight the significance of your research question.

3. Table 3 provides the good insights of the previous studies, however a critical review selected factors/measures is missing to answer the research question. Enhance this section of the manuscript, I recommend to categorize this section with respect to the selected factors/measures and impact on type of exoskeleton. This will help to highlight the strengths and weaknesses of previous research, and provide a clearer picture of how your study contributes to the field.

4. With respect to quality and productivity, commercial exoskeletons were not reviewed and ignored. In order to complete the comprehensive review, commercially available exoskeletons should also be included in the study, this may enhance the contribution of this study and helps to suggest conclusive recommendations.

5. The manuscript does include a brief discussion section, however recommendations are ignored or mixed with discussions. Comprehensive and conclusive findings and recommendations are missing, it is suggested to review all these sections and include additional headings to support the findings and arguments.

Reviewer #2: Overall, the presented paper is satisfactory. However, it is crucial to delve deeper into the factors that influence the quality and productivity outcome measures when utilizing an exoskeleton for occupational tasks. Additionally, it is recommended to review the excluded criteria in the screening step. Please revise them accordingly.

Kindly attach the checklist of questions pertaining to the inclusion criteria of the papers for reference. This will help ensure the selection process is comprehensive and accurate.

Lastly, to enhance the validity of the results, it is advisable to increase the number of papers included in the review. This will provide a more robust foundation for validating the findings.

6. PLOS authors have the option to publish the peer review history of their article (what does this mean?). If published, this will include your full peer review and any attached files.

Reviewer #1: **Yes: **Hammad Nazeer

Reviewer #2: No

---

## [Author Response · Author response to Decision Letter 0]

24 May 2023

We would like to thank both reviewers for their comments and the academic editor for the opportunity to improve this manuscript. We are pleased that reviewers found this manuscript to be well written and satisfactory. Please see our responses to each comment below. 

Reviewer #1: In this manuscript authors analyzed the effects of exoskeleton use on Quality and Productivity. For this purpose 15 critical studies were selected and reviewed for the factors/measures of endurance time, task completion time, number of errors, number of task cycles completed etc. Following are the concerns and suggestion to improve the manuscript

Authors Response: Thank you for your constructive feedback. We believe the following modifications significantly improved the manuscript. 

1. As per my understanding, the author's intension was to measure the quality and productivity of produced, developed and available exoskeleton in and out of lab applications. For this purpose keywords selected were not focused and had deficiencies, like human power augmentation, HCI based exoskeleton, BCI based exoskeletons, neurorobotics etc. These keywords may provide other critical studies for the mentioned topic.

Authors Response: Thank you for your comment. The intention of this systematic review was not to evaluate the quality and productivity of exoskeletons themselves. Instead, the purpose of this systematic review was to synthesize the current state of knowledge on the quality and productivity of workers and their work while wearing exoskeletons, as well as the economic implications of exoskeletons for occupational use. For example, we are interested in how the usage of exoskeletons could impact construction or assembly line workers’ quality and productivity of their work while wearing exoskeletons. 

In order to avoid possible confusion, we have added some clarification in the Introduction (lines 79-81).

2. The Introduction section is very brief and lacks a comprehensive review including introduction of exoskeleton, their types and applications. This will help to better contextualize your study and highlight the significance of your research question.

Authors Response: Thank you for your feedback. We have added additional information to further clarify the types and applications of exoskeletons (lines 42-43). We have also cited several studies to help readers obtain more in-depth information regarding different types of exoskeletons (lines 48-51).

3. Table 3 provides the good insights of the previous studies, however a critical review selected factors/measures is missing to answer the research question. Enhance this section of the manuscript, I recommend to categorize this section with respect to the selected factors/measures and impact on type of exoskeleton. This will help to highlight the strengths and weaknesses of previous research, and provide a clearer picture of how your study contributes to the field.

Authors Response: Thank you for your feedback. The purpose of our systematic review was to synthesize the current state of knowledge on the quality and productivity of workers and their work while wearing exoskeletons, as well as the economic implications of exoskeletons for occupational use. Performing a critical review of the measures on quality and productivity of exoskeleton products was beyond the scope of our systematic review. We understand that this might be a source of confusion and therefore, in response to your comment, we further clarified our research objective in the Introduction (lines 79-81).

4. With respect to quality and productivity, commercial exoskeletons were not reviewed and ignored. In order to complete the comprehensive review, commercially available exoskeletons should also be included in the study, this may enhance the contribution of this study and helps to suggest conclusive recommendations.

Authors Response: Thank you for your comment. We did not exclude commercial exoskeletons in our search strategy. “Exoskeleton” is one of the terms used in our literature search. Some of the papers that were retained for our review based on our systematic search and screening process, assessed the commercial exoskeletons designed for occupational application such as Laevo, HAL, BackX, and EksoWorks, and some assessed their own proprietary exoskeletons (Table 3). 

5. The manuscript does include a brief discussion section, however recommendations are ignored or mixed with discussions. Comprehensive and conclusive findings and recommendations are missing, it is suggested to review all these sections and include additional headings to support the findings and arguments.

Authors Response: Thank you for your feedback. In response to your comment, we have provided recommendations for future studies in the new Section 4.1 to better guide the reader. 

We clarified our conclusion in Section 4.3. Our systematic review provided mixed evidence regarding the effectiveness of exoskeleton based on quality and productivity measures. We also observed an absence of evidence on the economic impacts (e.g., cost, return on investment) of exoskeletons for occupational use. When selecting and adopting exoskeletons, task characteristics (e.g., required movements, workspace) may need to be considered to achieve the most effective outcomes for quality and productivity of workers and their work. More empirical studies are needed to improve our understanding on quality, productivity, and economic impacts of exoskeletons, including studies that consider sex and gender as well as studies that take place in actual work environments.

Reviewer #2: Overall, the presented paper is satisfactory. However, it is crucial to delve deeper into the factors that influence the quality and productivity outcome measures when utilizing an exoskeleton for occupational tasks. Additionally, it is recommended to review the excluded criteria in the screening step. Please revise them accordingly.

Authors Response: Thank you for your comment. We appreciate that you found this paper satisfactory. 

In this systematic review, our objective was to synthesize the state of knowledge on the quality and productivity of workers and their work while wearing exoskeletons, as well as the economic implications of exoskeletons for occupational use. Assessing the factors that influence the quality and productivity outcome measures was beyond the scope of our systematic review. In response to your comment, we have clarified our inclusion and exclusion criteria in Section 2.2.

Kindly attach the checklist of questions pertaining to the inclusion criteria of the papers for reference. This will help ensure the selection process is comprehensive and accurate.

Authors Response: Thank you for your feedback. We have attached our screening tool in S1 Appendix. It includes the checklist of questions pertaining to the inclusion criteria.

Lastly, to enhance the validity of the results, it is advisable to increase the number of papers included in the review. This will provide a more robust foundation for validating the findings.

Authors Response: Thank you for your comment. We followed the systematic review guidelines and PRISMA checklist to obtain the most relevant research articles pertaining to our research objective. Our search strategy included six large databases (i.e., PubMed, Medline, Embase, Business Source Complete, IEEE Xplore, and Scopus) and we identified 6,722 articles in our initial search. We had to exclude 670 duplicates as well as 6,037 articles because they did not meet the inclusion criteria.

---

## [Decision Letter · Decision Letter 1]

14 Jun 2023

Quality, productivity, and economic implications of exoskeletons for occupational use: A systematic review

PONE-D-23-09205R1

Dear Dr. Yazdani,

We’re pleased to inform you that your manuscript has been judged scientifically suitable for publication and will be formally accepted for publication once it meets all outstanding technical requirements.

Kind regards,

Noman Naseer, PhD

Academic Editor

PLOS ONE

Additional Editor Comments (optional):

The comments have been adequetly addressed. The paper is in acceptable form now.

Reviewers' comments:

Reviewer's Responses to Questions

**Comments to the Author**

1. If the authors have adequately addressed your comments raised in a previous round of review and you feel that this manuscript is now acceptable for publication, you may indicate that here to bypass the “Comments to the Author” section, enter your conflict of interest statement in the “Confidential to Editor” section, and submit your "Accept" recommendation.

Reviewer #1: All comments have been addressed

Reviewer #2: All comments have been addressed

2. Is the manuscript technically sound, and do the data support the conclusions?

Reviewer #1: Yes

Reviewer #2: Yes

3. Has the statistical analysis been performed appropriately and rigorously? 

Reviewer #1: Yes

Reviewer #2: N/A

4. Have the authors made all data underlying the findings in their manuscript fully available?

Reviewer #1: Yes

Reviewer #2: Yes

5. Is the manuscript presented in an intelligible fashion and written in standard English?

Reviewer #1: Yes

Reviewer #2: Yes

6. Review Comments to the Author

Reviewer #1: The Authors have addressed all of my concerns with the original manuscript. The revised manuscript is ready for publication.

Reviewer #2: I would like to express my gratitude for submitting the revised version of your manuscript in response to the comments and suggestions provided by the reviewers, including my own. Thank you for taking the time and effort to address the concerns raised and for providing detailed clarifications in the revised manuscript.

7. PLOS authors have the option to publish the peer review history of their article (what does this mean?). If published, this will include your full peer review and any attached files.

Reviewer #1: **Yes: **Hammad Nazeer

Reviewer #2: No

---

## [Editor Report · Acceptance letter]

19 Jun 2023

PONE-D-23-09205R1 

Quality, productivity, and economic implications of exoskeletons for occupational use: A systematic review 

Dear Dr. Yazdani:

I'm pleased to inform you that your manuscript has been deemed suitable for publication in PLOS ONE. Congratulations! Your manuscript is now with our production department. 

Kind regards, 

on behalf of

Dr. Noman Naseer 

Academic Editor

PLOS ONE